# Self-Assembled Amphiphilic Chitosan Nanomicelles: Synthesis, Characterization and Antibacterial Activity

**DOI:** 10.3390/biom13111595

**Published:** 2023-10-30

**Authors:** Yi Qi, Qizhou Chen, Xiaofen Cai, Lifen Liu, Yuwei Jiang, Xufeng Zhu, Zhicheng Huang, Kefeng Wu, Hui Luo, Qianqian Ouyang

**Affiliations:** 1Marine Biomedical Research Institution, Guangdong Medical University, Zhanjiang 524023, China; qiyi7272@gdmu.edu.cn (Y.Q.); sj13423409060@gmail.com (Q.C.); 13413676045@163.com (X.C.); 15125044514@163.com (L.L.); jyw20000215@163.com (Y.J.); xufeng910730@126.com (X.Z.); 498556764@163.com (Z.H.); luohui@gdmu.edu.cn (H.L.); 2The Marine Biomedical Research Institute of Guangdong Zhanjiang, Zhanjiang 524023, China; 3The Key Lab of Zhanjiang for R&D Marine Microbial Resources in the Beibu Gulf Rim, Marine Biomedical Research Institute, Guangdong Medical University, Zhanjiang 524023, China

**Keywords:** deoxycholic acid, succinic anhydride, amphiphilic chitosan, nanomicelles, antibacterial activity

## Abstract

Although amphiphilic chitosan has been widely studied as a drug carrier for drug delivery, fewer studies have been conducted on the antimicrobial activity of amphiphilic chitosan. In this study, we successfully synthesized deoxycholic acid-modified chitosan (CS-DA) by grafting deoxycholic acid (DA) onto chitosan C_2_-NH_2_, followed by grafting succinic anhydride, to prepare a novel amphiphilic chitosan (CS-DA-SA). The substitution degree was 23.93% for deoxycholic acid and 29.25% for succinic anhydride. Both CS-DA and CS-DA-SA showed good blood compatibility. Notably, the synthesized CS-DA-SA can self-assemble to form nanomicelles at low concentrations in an aqueous environment. The results of CS, CS-DA, and CS-DA-SA against *Escherichia coli* and *Staphylococcus aureus* showed that CS-DA and CS-DA-SA exhibited stronger antimicrobial effects than CS. CS-DA-SA may exert its antimicrobial effect by disrupting cell membranes or forming a membrane on the cell surface. Overall, the novel CS-DA-SA biomaterials have a promising future in antibacterial therapy.

## 1. Introduction

In recent years, due to the misuse of antibiotics by humans, many microorganisms have become highly resistant to commonly used antibiotics, which has led to increased morbidity and mortality from infectious diseases and has also become a serious public health problem worldwide [1,2]. Due to the long time, high cost, and little effect required to discover new antimicrobial agents from natural products, increasing attention is being paid to the use of synthetic compounds with antimicrobial activity. Natural antimicrobial polymers are biocompatible, biodegradable, safe, and non-toxic, and can exert antibacterial or bactericidal effects through their chemical structures, making them a new generation of antimicrobial agents [3]. Currently, nanomaterials have developed as promising materials in various fields of science and technology. In particular, they have become the most widely used materials in medical practice [4]. Nanomaterials have been widely used in wastewater treatment, antimicrobials, photothermal therapy, biomedical imaging, and cancer treatment [5,6]. To address the problem of antibiotic resistance, more and more researchers are working on developing alternatives to traditional antibiotics. Compared with antibiotics, nanomaterials can produce synergistic antimicrobial effects through multiple mechanisms of action and can act on specific targets within the bacterial cell, reducing the risk of resistance development [7]. Nanomaterials have the advantages of broad-spectrum antibacterials, high efficiency, and low toxicity, and can play an antibacterial role by destroying bacterial cell membranes or interfering with bacterial physiological activity [8]. For example, A.F. Jafarova et al. [9] prepared environmentally friendly and non-toxic silver nanoparticles using biological methods and investigated their antibacterial activity against Bacillus subtilis and *Staphylococcus aureus*, and the results showed that silver nanoparticles with sizes in the range of 50–100 nm can solubilize the bacteria and produce a good antibacterial effect.

Chitosan is the only known and abundant natural basic cationic polymer [10]. It has positively charged amino groups that bind to their anionic counterparts in the mucous membrane layer and can be administered through the gastrointestinal tract, skin, and nasal passages [11]. Chitosan has good biodegradability, biocompatibility, and cellular affinity [12,13]. Chitosan has rich pharmacological effects such as antibacterial, hemostatic, antiviral, and inhibitory effects on a wide range of bacteria and fungi [14,15,16]. Studies have shown that the antibacterial principle of chitosan may result from electrostatic interactions between its positively charged amino group and negative charges present on microbial membranes, resulting in antibacterial effects [17]. However, chitosan is poorly water-soluble and has a weak positive charge; thus, the antimicrobial activity is weak [18,19]. The antimicrobial properties of chitosan are associated with various factors, including molecular weight, pH, temperature, and the extent of deacetylation [20], and its antimicrobial activity is unstable, so its application in the antimicrobial field is limited. The chemical properties of -NH_2_ and -OH of chitosan molecular chains are active and can be chemically modified to synthesize chitosan derivatives [21]. More and more studies have shown that chitosan derivatives synthesized by the chemical modification of chitosan can enhance its antimicrobial activity [22,23]. For example, Fengling Tang et al. [24] used succinic anhydride and basic chitosan as raw materials for the synthesis of N-succinyl chitosan (NSC), a water-soluble derivative of chitosan, finding that CS solubility was markedly enhanced by NSC, together with excellent antibacterial properties and antimicrobial activity. NSC reduced bacterial infection and promoted the formation of granulation tissue and epithelialization in rabbit skin wounds, with markedly shorter healing times relative to CS. Nuraziemah Ahmad et al. [25] synthesized amphiphilic chitosan (CH-Arg-OA) by grafting arginine and oleic acid through a carbodiimide-mediated reaction and investigated its antimicrobial activity, and the resultant CH-Arg-OA showed significant antibacterial action, particularly against gram-negative bacteria. These effects were due to disruption of the outer membranes of the bacteria, minimizing the possibility of bacterial drug resistance.

The structure of amphiphilic chitosan has both hydrophilic and hydrophobic groups, which can self-assemble to form super-stable nanomicelles when dissolved in water [26,27], leading to increased stability of the antimicrobial action of the amphiphilic chitosan nanomicelles. The present study aimed to improve both the antibacterial properties and stability of chitosan and thus synthesized amphiphilic chitosan using the hydrophobic modification of chitosan with deoxycholic acid and the hydrophilic modification of chitosan with succinic anhydride. Deoxycholic acid is often used as a hydrophobic group for the hydrophobic modification of amphiphilic polymers, and has strong antimicrobial activity [28]. Succinic anhydride can be hydrolyzed to succinic acid, which is strongly acidic, allowing high nanocellulose yields to be achieved [29]. Succinic anhydride is commonly used for the hydrophilic modification of chitosan, and its synthetic product, N-succinyl chitosan, also has good antibacterial activity [24]. The CS-DA-SA structure was characterized by nuclear magnetic resonance hydrogen spectroscopy, Fourier-transform infrared spectroscopy, and X-ray diffraction. Ultrasonication reduces aggregation and releases individual particles, reducing viscosity between materials [30]. Ultrasonic self-assembly was used to prepare the amphiphilic chitosan nanomicelles (CS-DA-SA-NMs), and their sizes, zeta potentials, and microscopic structures were investigated. The antibacterial properties of CS, CS-DA, and CS-DA-SA were compared and analyzed, and the possible mechanisms of their antibacterial effects were discussed. In this paper, novel amphiphilic polymeric nanomicelles were prepared to improve the antibacterial activity and stability of chitosan.

## 2. Experimental

### 2.1. Materials

Chitosan (CS, viscosity < 200 mPa.s, 97.8% deacetylated), N-acetyl-L-cysteine (NAC), 1-ethyl-3(3-dimethylaminopropyl) carbodiimide hydrochloride (EDAC), deoxycholic acid (DA), and succinic anhydride (SA) were purchased from Macklin (Shanghai, China). Both *E. coli* (CMCC(B) 44103) and *S. aureus* (ATCC 25923) were from the China Center of Industrial Culture Collection, Beijing, China. In addition, methanol, ammonia, glacial acetic acid, ethanol, acetone, and DMSO were from Guanghua Technology Co., Ltd. (Shantou, China). The reagents and chemicals were all of analytical grade and were not further purified. All experiments used ultrapure water.

### 2.2. Synthesis of Chitosan Derivatives

#### 2.2.1. CS-DA Synthesis

Referring to the modified method in the literature [31], a clean 250 mL beaker was taken, to which 75 mL of 1% acetic acid solution and 0.5 g chitosan were added sequentially and stirred to dissolve chitosan completely. Take another beaker and add 75 mL of methanol and 1.217 g of deoxycholic acid, then add 0.296 g of EDAC and 0.538 g of NHS, respectively, and stir for 2 h for activation. After activation, transfer it into a chitosan solution. If milky turbidity appears, add some methanol to dissolve it, and stir the reaction for 24 h at ambient temperature. Then, add ammonia while stirring until the product is completely at the end of the reaction. Let it stand for 5 min and then filter. Wash with anhydrous ethanol and filter and repeat 3 times. Dry under a vacuum to obtain deoxycholic acid-modified chitosan (CS-DA). Details of the synthesis are shown in Appendix A.

#### 2.2.2. CS-DA-SA Synthesis

Referring to the modified method in the literature [32], 0.25 g CS-DA was weighed, followed by dissolution in 50 mL of 1% acetic acid, and another 0.166 g succinic anhydride was weighed and dissolved in 20 mL of acetone, followed by a dropwise addition of the succinic anhydride/acetone solution to CS-DA with constant stirring for 4 h at 40 °C. After the reaction, excess acetone was added until the product was complete. After the reaction, add excess acetone until the product is completely analyzed, and then filter and wash the solid phase with 70%, 80%, and 100% acetone, respectively, and dry it under a vacuum for 24 h. The amphiphilic chitosan (CS-DA-SA) is obtained. Details of the synthesis are shown in Appendix A.

### 2.3. Identification and Characterization of Compounds

FTIR spectra of the chitosan and derivatives were obtained using the potassium bromide compression method. ^1^H NMR spectra at 400 MHz were obtained using a Bruker AVANCE NEO 400 MHz spectrometer. CS, CS-DA, and CS-DA-SA were dissolved in a mixture of D_2_O and C_2_D_4_O_2_. The crystallization behavior of chitosan and its derivatives were analyzed by a Rigaku Smart Lab 9 kW X-ray diffractometer with scanning angles of 10°–80°.

### 2.4. Determination of the Grafting Rate

#### 2.4.1. DA Grafting Rate

The rate of deoxycholic acid grafting on chitosan can be quantified by reacting the chitosan-free amino group with ninhydrin [33]. Five chitosan solutions were prepared at concentrations of 0.25, 0.5, 1, 2, and 4 mg/mL, respectively. Assuming that the concentration of 4 mg/mL chitosan has 100% free amino acid, the other four chitosan solutions have 50%, 25%, 12.5%, and 6.25% free amino acid (C), respectively. Take 0.5 mL of each of the above chitosan solutions, add 0.5 mL of an acetate buffer (pH 5.5) and 2 mL of aqueous ninhydrin (5 mg of ninhydrin, dissolved in 100 mL of ethanol) dropwise, heat mixed solutions in a water bath at 90 °C, and stir magnetically for 15 min. Absorbances (A) at 570 nm were determined on a UV spectrophotometer. Linear regression of A and C was determined as C = aA + b. The absorbance of a 2 mg/mL solution of a CS-DA solution was measured. From the linear regression equation C = aA + b, the corresponding amount of free amino acid could be calculated, and the grafting rate of CS-DA is 100%—C.

#### 2.4.2. SA Grafting Rate

The C, H, and N contents in CS-DA and CS-DA-SA were determined by the Dumas combustion method using an Elementar vario EL cube organic elemental analyzer. The amount of SA substitution was evaluated using Equation (1).
(1)$$\mathrm{DS}=\frac{{C/N}_{CS-DA-SA}-{C/N}_{CS-DA}}{{C/N}_{SA-DA-GCA}-{C/N}_{GCA}}$$
where C/N_SA-DA-GCA_ is the C/N ratio of the sugar unit after substitution with succinic anhydride. C/N_GCA_ is the C/N ratio of the deacetylated sugar unit.

### 2.5. Preparation of Amphiphilic Chitosan Nanomicelles

These were prepared using an ultrasonic self-assembly method [34]. One milligram of CS-DA-SA was dissolved in 10 mL of 1% acetic acid, followed by sonication for 10 min at room temperature, to prepare the CS-DA-SA-NMs.

### 2.6. Measurement of Particle Sizes and Zeta Potentials 

Particle sizes, zeta potentials, and their distribution in the nanomicelles were evaluated using a Malvern nanoparticle size and potential analyzer (DLS). The CS-DA-SA-NMs were placed on the copper network and stained with a phosphotungstic acid negative staining solution for 2 min. The morphology of amphiphilic chitosan nanomicelles was examined under transmission electron microscopy (TEM).

### 2.7. Calculation of the Critical Micellar Concentration (CMC) of CS-DA-SA-NMs

The CMC values were evaluated using pyrene as a molecular probe [35]. A total of 4 mg of CS-DA-SA were dissolved in 10 mL of 1% acetic acid, followed by sonication for 10 min at room temperature, to yield 0.4 mg/mL of CS-DA-SA-NMs. This solution was diluted to varying concentrations of nanomicelles (0.2, 0.1, 0.05, 0.025, 0.025, 0.0125, 0.00625, and 0.003125 mg/mL). Fifty microliters of 6 × 10^−5^ mg/mL pyrene/acetone solution were added to 5 mL of each of these dilutions. They were mixed and shaken well, followed by sonication for 15 min, heating (50 °C, 2 h), and incubation overnight at room temperature to reach equilibrium. The fluorescence spectra of different concentration polymer micelles were scanned using a fluorescence spectrophotometer with excitation at 339 nm. They were scanned between 360 and 450 nm using 5.0 nm slit widths for both excitation and emission.

### 2.8. Hemocompatibility of CS-DA-SA-NMs

The toxicity of nanomicelles was tested calorimetrically by hemoglobin release from erythrocytes, as described [36]. Specifically, blood was collected from SD rat hearts into heparinized tubes and centrifuged (3000 rpm, 10 min), followed by three saline washes and a resuspension of the erythrocytes to 2% (*v*/*v*) in saline. Two milliliters of the suspension were added to 2.0 mL of the CS-DA and CS-DA-SA solutions, followed by incubation (1 h, 37 °C) and centrifugation (3000 rpm, 15 min). The absorbances of the supernatants were determined at 540 nm using normal saline as the negative control (0% hemolysis) and ultrapure water as the positive control (100% hemolysis). The hemolysis percentage was determined using Equation (2).
(2)$$\mathrm{Hemolysis}\;(\%)=\frac{A_{Sample}-A_{Saline}}{A_{Water}-A_{Saline}}\times 100\%$$

### 2.9. Assays for Antimicrobial Activity Evaluation

#### 2.9.1. Paper Diffusion

The described protocol [37] was used to assess the inhibition circles of CS, CS-DA, and CS-DA-SA. CS, CS-DA, and CS-DA-SA solutions of 0.125, 0.25, 0.5, 1, and 2 mg/mL were prepared using 0.2% acetic acid as solvent. *Escherichia coli* and *Staphylococcus aureus* were diluted to 10^6^ CFU/mL using saline, and 1 mL of bacterial solution was added to the Petri dishes. Then, an LB agar medium was poured, and mixed well. After the addition of sterile 6 mm blank drug-sensitive tablets to the solid agar, the drug solution (20 μL) was included with the drug-sensitive tablets. A total of 0.2% acetic acid and levofloxacin were the negative and positive controls, respectively, and the drug solution was diffused for 2 h in a refrigerator at 4 °C, followed by incubation for 24 h at 37 °C. The diameters of the inhibition zones (in mm) were determined by electronic vernier calipers. The experiment was repeated three times.

#### 2.9.2. Measurement of Minimum Inhibitory Concentrations

These were measured for chitosan and chitosan derivatives against *E. coli* and *S. aureus* using the broth two-fold dilution method (the concentration corresponding to 90% inhibition). The rates of inhibition were calculated by counting the number of bacterial growth colonies in the nutrient agar dilution method. First, the prepared samples (dissolved in 0.2% acetic acid to make a mother liquor concentration of 2 mg/mL) were included with the bacterial culture solution (10^6^ CFU/mL) so that the concentration of each test tube was 0, 0.0078125, 0.015625, 0.03125, 0.0625, or 0 mg/mL. Three parallel groups were set up by adding 4 μL of *Escherichia coli* suspension or *Staphylococcus aureus* suspension to each of the seven concentration gradients in 96-well plates, followed by incubation for 12 h at 37 °C. A total of 100 μL of bacterial cultures were removed and serially diluted with sterile saline to the predetermined concentrations. Finally, 1 mL of a bacterial solution was added to the culture dish, and then the LB agar medium was poured, mixed well, and incubated for 24 h at 37 °C. Live microbial colonies were obtained and counted.

#### 2.9.3. Analysis of the Integrity of Bacterial Membranes

The integrity of the membranes was assessed by measuring DNA and RNA release from bacterial cells [38]. The method was modified according to a previous study [39]. The experiments were grouped into the control, CS (2 MIC), CS-DA (2 MIC), and CS-DA-SA (2 MIC) groups. The final concentration of the bacterial suspension was adjusted to 10^7^ CFU/mL, and the test organisms were mixed with an equal volume of sample, followed by incubation for 12 h at 37 °C with shaking at 150 rpm/min. The suspensions were centrifuged (5000 rpm, 5 min, 4 °C), and absorbances at 260 nm were measured in the supernatants against an enzyme standard.

#### 2.9.4. Effects of CS-DA-SA on Bacterial Morphology

The effects of CS, CS-DA, and CS-DA-SA on the morphology of *E. coli* and *S. aureus* were observed by SEM. As described [40] with some modifications, 3 mL of bacteria in stable or late-logarithmic growth were centrifuged (8000 rpm, 5 min), and the pelleted cells were fixed in 40 volumes of 2.5% glutaraldehyde at 4 °C for a minimum of 2 h. The bacteria were washed several times in a phosphate buffer and dehydrated (50, 70, and 90% ethanol). They were then resuspended in an ethanol tert-butanol solution (1:1, *v*/*v*) for 20 min. Finally, they were replaced twice with 100% tert-butanol, 20 min each time. After replacement, the samples were subjected to freeze-drying treatment, and the dried samples were observed and photographed using a scanning electron microscope (SEM) after being sprayed by an ion-sputtering apparatus.

#### 2.9.5. Data Processing

OriginPro 2021 (OriginLab, Northampton, MA, USA) software was used for statistical analysis, and all data were expressed as $\bar{x}$ ± s. One-way ANOVA and LSD were used to test the one-way variance; * *p* < 0.05 was considered a statistically significant difference and ** *p* < 0.01 was considered a highly significant difference.

## 3. Results and Discussion

### 3.1. Characterization of Chitosan Derivatives

The chitosan derivatives (CS-DA-SA) were successfully prepared (Appendix A). The structure of CS-DA-SA was evaluated and verified by FTIR, ^1^H NMR, and XRD spectra (Appendix A). The amphiphilic chitosan was successfully synthesized by introducing hydrophilic and hydrophobic groups into the C_2_-NH_2_ position of chitosan. The standard curve of the chitosan-free amino amount is shown in Figure 1, and the rate of deoxycholic acid grafting was calculated to be 23.93% by substituting it into the equation. The results of the elemental analysis of CS-DA and CS-DA-SA are presented in Table 1, and the grafting rate of succinic anhydride was measured as 29.25%.

### 3.2. Measurement of the CMC

It is important that the concentration of the amphiphilic chitosan solution is higher or equal to the CMC to allow self-assembly of the polymers and their aggregation within shell–core structures. Pyrene is sensitive to microenvironmental polarities, and the ratio of the intensities of the first peak of its fluorescence spectrum (I_1_, 373 nm) to the third peak (I_3_, 383 nm) is used as a measure of these variations [41]. The CMC values of the amphiphilic chitosan nanomicelles are illustrated in Figure 2. The CMC of CS-DA-SA was found to be 0.05 mg/mL, indicating that CS-DA-SA can self-assemble to form nanomicelles at lower concentrations and more stable self-aggregation under dilution conditions [41]. The CMC values of CS-DA-SA were lower than similar amphiphilic chitosan nanocellulose systems, such as deoxycholic acid-O-carboxymethylated chitosan–folic acid affixes [42] and novel amphiphilic chitosan derivatives with hydrophilic carboxyl groups and hydrophobic hexadecyl groups [41]. 

### 3.3. Particle Sizes, Zeta Potentials, and Morphological Analysis of Chitosan Derivatives

The addition of cationic or hydrophobic groups to chitosan usually leads to the acquisition of different physicochemical properties, including the capacity for nanoparticle self-assembly [25]. TEM was used for the morphological evaluation of the CS-DA-SA nanomicelles. As seen in Figure 3, the CS-DA-SA nanomicelles appeared spherical or quasi-spherical in shape with an even distribution. The self-assembly of CS-DA-SA resulted in nanoparticle diameters between 20 and 50 nm. However, shadowy structures were seen surrounding the particles in the TEM images. These may have been the result of flocculation during dehydration when the samples were prepared for TEM [43]. 

Size and zeta potential are the basic characteristic parameters of nanosuspensions [44]. Table 2 shows the characteristics of CS-DA and CS-DA-SA-NMs. The diameter of CS-DA-SA-NMs nanomicelles was 827.47 ± 39.43 nm with a polydispersity index (PDI) of 0.93 ± 0.08, indicating the narrow size distribution of the system. The zeta potential is a measure of surface charge and can significantly influence particle stability in solution due to inter-particle electrostatic repulsive forces [45]. The suspensions were considered stable when the zeta potentials were less than −15 mV or higher than +15 mV [46]. Both CS-DA and CS-DA-SA-NMs showed potentials greater than 15 mV, and CS-DA-SA-NMs were significantly higher. This indicates that CS-DA-SA-NMs are more stable and improve the stability of CS-DA.

### 3.4. Hemocompatibility

Interactions between the free amino moieties of chitosan and cells or proteins in the circulation can lead to thrombosis or hemolytic reactions [47]. The contact of polymers with blood can be harmful or even instantly fatal [48], so it is important to determine the hemocompatibility of novel materials. Hemolysis is considered to be a very simple and reliable measurement to assess the hemocompatibility of materials. CS-DA and CS-DA-SA-NMs were incubated with erythrocytes separately to observe their effects on erythrocytes. The hemolysis rates were found to be 0.44 ± 0.09% for CS-DA and 0.16 ± 0.13% for CS-DA-SA-NMs, both of which are below the international standard of 5%, demonstrating good hemocompatibility in both. Wei-Yan Quan et al. [49] modified chitosan by an acylation reaction and grafted 18β-glycyrrhetinic acid and sialic acid, respectively, to synthesize amphiphilic chitosan, and found that the modified chitosan and the blood compatibility of sugar derivatives were significantly improved.

### 3.5. Antimicrobial Activity of CS-DA-SA

#### 3.5.1. Agar Paper Sheet Diffusion

Previous studies have shown [22] that zeta potential has a strong influence on the antibacterial properties of nanoparticles, suggesting that these properties may result from interactions between the positively charged amino groups of chitosan with negative charges on the bacterial membrane. In this study, the inhibitory effects of CS, CS-DA, and CS-DA-SA on gram-negative (*E. coli*) and gram-positive (*S. aureus*) bacteria were investigated by paper diffusion method. The results of agar paper diffusion tests for the antibacterial activity of chitosan and its derivatives are presented in Table 3. It was found that varying concentrations of CS, CS-DA, and CS-DA-SA could inhibit the growth of both *E. coli* and *S. aureus*, with relatively similar inhibitory effects. Between 0.5 and 2 mg/mL, the inhibition effects of CS, CS-DA, and CS-DA-SA showed a concentration–dependent relationship. The inhibition effect of CS-DA-SA was significantly enhanced compared with CS and CS-DA. The possible reason is that the synthesized CS-DA-SA has an increased zeta potential value and an increased number of positive charges, thus enhancing its antibacterial ability. It is also possible that the antibacterial properties of CS were enhanced by the presence of succinic anhydride [22]. Jinping Cai et al. [50] synthesized a novel amphiphilic chitosan derivative, N-benzoyl-O-acetyl-chitosan (BACS). It was found that BACS also exhibited stronger antibacterial activity against *E. coli* and *S. aureus* compared to CS.

#### 3.5.2. Minimum Bacterial Inhibition Concentration

To determine the minimum concentration that can completely block bacterial growth, the minimum inhibitory concentrations of chitosan and its derivatives were determined after treatment of *E. coli* and *S. aureus* by the two-fold dilution method. The findings are illustrated in Table 4. Our results showed that the minimum inhibitory concentrations of CS-DA and CS-DA-SA-NMs were significantly lower and smaller compared to CS, indicating that CS-DA and CS-DA-SA improved the antibacterial activity of CS and had better antibacterial effects.

#### 3.5.3. Microbial Growth Kinetics

To further verify the antimicrobial properties of chitosan and its derivatives, their effects on bacterial growth were evaluated using growth curves. The growth curves of the tested strains in the culture medium of chitosan and its derivatives are provided in Figure 4, indicating that the growth rates of both bacteria in the blank group showed an “S”-like curve. CS, CS-DA, and CS-DA-SA were effective in slowing bacterial growth in a short period (1–2 h), and the OD_600_ values in the culture medium were small and changed very little after 2 h, indicating that they all had strong antibacterial effects. Overall, the OD_600_ values of the bacterial cultures were ranked as CS > CS-DA > CS-DA-SA, indicating that the modified chitosan derivatives enhanced the antimicrobial properties of chitosan, which was consistent with the results of paper diffusion experiments.

#### 3.5.4. Cell Membrane Integrity

The cell membrane is important for bacterial growth. Antibacterial agents can affect cell proliferation and differentiation, damage cell membranes, and leak small and macromolecular substances in cells [39,51]. The effects of chitosan and its derivatives on membrane integrity were revealed by measuring the amount of genetic material permeated from the cell membrane (Figure 5). It was found that compared with the blank group, both the CS and CS-DA-SA groups significantly increased the amount of DNA and RNA leakage of the intracellular genetic material of *Escherichia coli* and *Staphylococcus aureus* (*p* < 0.01). This indicated that CS and CS-DA-SA exerted an antibacterial effect by disrupting the integrity of the cell membrane. This is consistent with the results of Nuraziemah Ahmad et al. [25].

#### 3.5.5. Scanning Electron Microscopy

Morphological changes in bacteria induced by chitosan and its derivatives were evaluated by SEM (Figure 6). The SEM maps of *E. coli* showed that the control bacteria were about 2 μm long, with bacillary form and smooth surface. The cell surface of the CS group was corrugated, and there were more pits inside the cells. The cell surface of the CS-DA and CS-DA-SA groups appeared depressed and distorted in shape. The cells of the CS-DA-SA group appeared crumpled, and some bacteria ruptured, probably because it could disrupt the bacterial cell membrane and prompt *E. coli* bacteria to crumple, lyse, and die. Shiqi He et al. [52] synthesized an amphiphilic polysiloxane ammonium salt (PDMS-g-AH), which also has a better inhibitory effect on *E. coli*, and the micelles formed by PDMS-g-AH have a high zeta potential, and the high positive charge density makes its binding ability to the bacterial cell membrane stronger so that it can easily interact with the microbial cell membrane. SEM images of *S. aureus* showed that the control group cells were about 0.8 μm in length, spherical in shape, smooth in surface, and arranged in grape bunches. The CS group cells were irregular in size, and more cells underwent rupture to produce fragments, probably due to disrupted cells or cell membrane dysfunction [53]. Most of the cells in the CS-DA group were divided and severely distorted, producing a large number of fragments. In the CS -DA-SA group of cells, some of the cells are divided to produce fragments. Another part of the cells was spherical and had a rough surface. The underlying antimicrobial mechanism may be the prevention of nutrient or oxygen entry into the bacterium due to membrane formation on the surface of the cell, thus inhibiting growth [54].

## 4. Conclusions

This study synthesized novel amphiphilic chitosan by chemically modifying its structure with chitosan as a raw material, grafting the hydrophobic group DA and hydrophilic group SA at C_2_-NH_2_, respectively, and characterizing its structure. Ultrasonic self-assembly was used for the preparation of amphiphilic chitosan nanomicelles. The CMC of CS-DA-SA-NMs was found to be 0.05 mg/mL, indicating that they can form micelles at low concentrations. The particle size of CS-DA-SA-NMs was 827.47 ± 39.43 nm, the zeta potential was +32 ± 0.28 mv, and the morphology was spherical. It was found that the modified chitosan derivatives, CS-DA and CS-DA-SA, both had strong antibacterial activity. The antibacterial activity of CS-DA and CS-DA-SA was significantly higher compared to CS, and the MIC values of CS-DA and CS-DA-SA were lower. CS-DA-SA may exert its antibacterial effect by disrupting the cell membrane or possibly forming a membrane on the cell surface. In conclusion, these results provide a theoretical foundation for the development of natural antimicrobial polymers into antimicrobial agents with improved and stable antimicrobial activity.

## Figures and Tables

**Figure 1 biomolecules-13-01595-f001:**
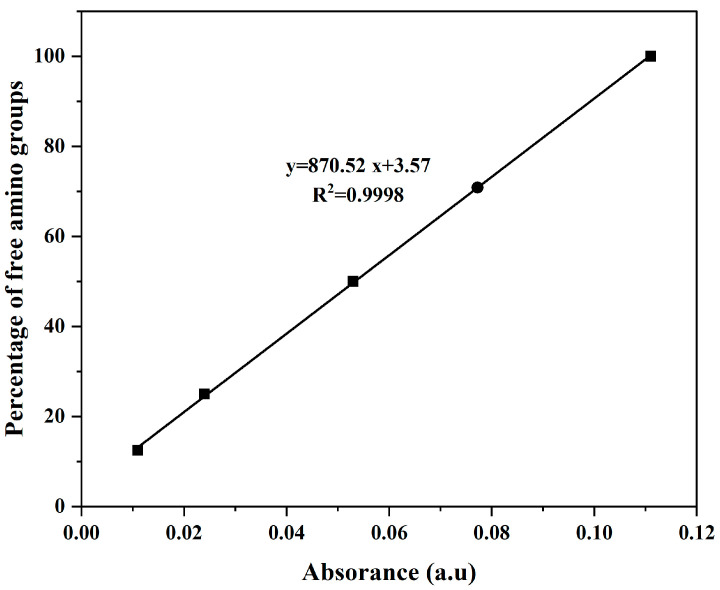
Standard curve of the free amino acid amount of chitosan.

**Figure 2 biomolecules-13-01595-f002:**
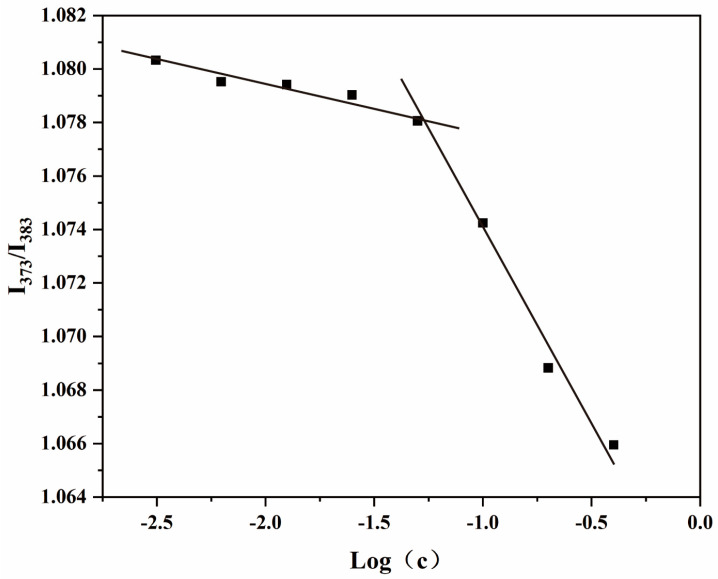
Critical micellar concentration of CS-DA-SA.

**Figure 3 biomolecules-13-01595-f003:**
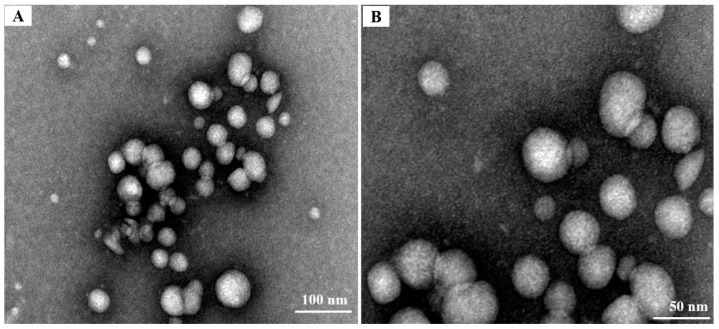
(**A**) TEM images of CS-DA-SA at a scale of 100 nm. (**B**): TEM images of CS-DA-SA at a scale of 50 nm.

**Figure 4 biomolecules-13-01595-f004:**
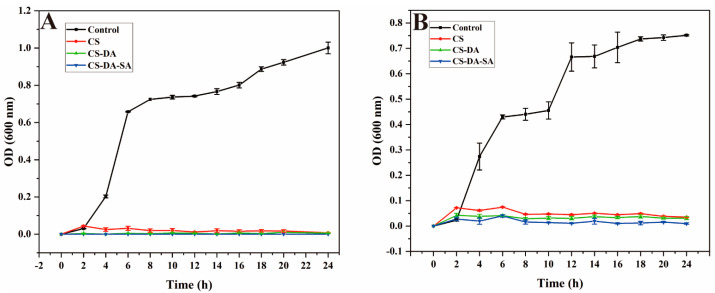
(**A**) Growth curves of *E. coli* after exposure to the control, CS, CS-DA, CS-DA-SA; (**B**): growth curves of *S. aureus* after exposure to the control, CS, CS-DA, CS-DA-SA.

**Figure 5 biomolecules-13-01595-f005:**
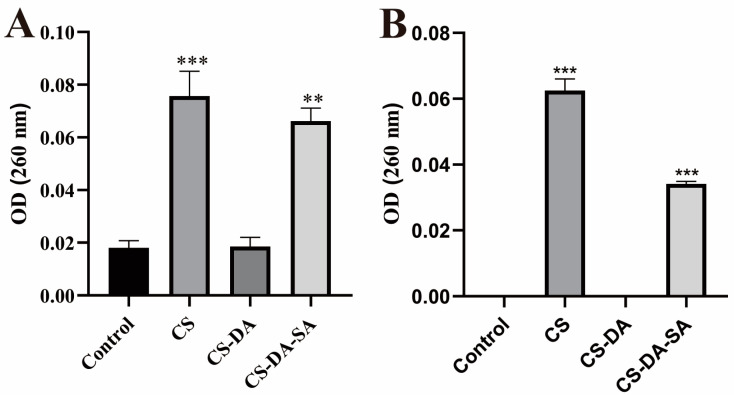
Absorption values at 260 nm for the release of nucleic acids (DNA and RNA) from *E. coli* (**A**) and *S. aureus* (**B**) treated in the control, CS, CS-DA, and CS-DA-SA groups. Versus the control group, ** *p* < 0.01, *** *p* < 0.001.

**Figure 6 biomolecules-13-01595-f006:**
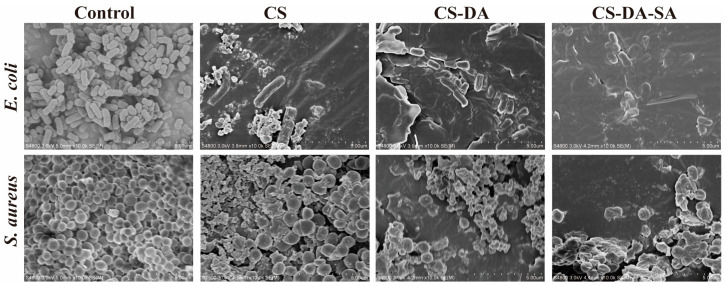
SEM morphology of *E. coli* and *S. aureus* in the control, CS, CS-DA, and CS-DA-SA groups at a scale of 5 μm.

**Table 1 biomolecules-13-01595-t001:** Elemental analysis table of CS-DA and CS-DA-SA.

Name	Element C (%)	Element H (%)	Element N (%)
CS-DA	42.64	6.99	7.4
CS-DA-SA	34.91	6.44	5.16

**Table 2 biomolecules-13-01595-t002:** Zeta potentials and sizes of the synthesized CS-DA and CS-DA-SA.

Samples	Size (nm)	PDI	Zeta (mV)
CS-DA	1658.33 ± 192.02	0.74 ± 0.17	+25.97 ± 0.33
CS-DA-SA	827.47 ± 39.43	0.93 ± 0.08	+32 ± 0.28

PDI is the polydispersity index.

**Table 3 biomolecules-13-01595-t003:** Antibacterial activities of CS-DA against *E.coli* and *S.aureus*.

Strain	Concentration (mg/mL)	CS	CS-DA	CS-DA-SA	0.2% CH_3_COOH	Levofloxacin
		Size of Inhibition Zone (mm)				
*S.aureus*	0.125	7.63 ± 0.21	7.47 ± 0.46	7.67 ± 0.31	0	9.90 ± 0.56
	0.25	8.53 ± 0.32	9.40 ± 0.26	7.43 ± 0.060		
	0.5	7.47 ± 0.21	8.33 ± 0.50	7.60 ± 0.26		
	1	7.67 ± 0.46	9.30 ± 0.36	9.47 ± 0.25		
	2	7.80 ± 0.20	8.80 ± 0.20	10.33 ± 0.32		
*E.coli*	0.125	7.03 ± 0.15	7.07 ± 0.23	8.03 ± 0.40	0	10.57 ± 0.38
	0.25	7.17 ± 0.15	7.40 ± 0.10	8.33 ± 0.06		
	0.5	7.13 ± 0.15	8.13 ± 0.38	8.03 ± 0.15		
	1	7.20 ± 0.20	8.43 ± 0.40	9.87 ± 0.42		
	2	7.33 ± 0.31	8.47 ± 0.25	10.80 ± 0.44		

**Table 4 biomolecules-13-01595-t004:** MIC values of CS-DA and CS-DA-SA.

Strain	CS (mg/mL)	CS-DA (mg/mL)	CS-DA-SA (mg/mL)
*S.aureus*	0.125	0.0078125	0.015625
*E.coli*	0.125	0.0078125	0.015625

## Data Availability

The data are available from the corresponding author upon reasonable request.

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
