# Peer review of "Self-Assembled Amphiphilic Chitosan Nanomicelles: Synthesis, Characterization and Antibacterial Activity"

_biomolecules, 2023, doi:10.3390/biom13111595_

Round 1
Reviewer 1 Report
Comments and Suggestions for Authors
The article fits the journal however there are some necessary corrections:
a) English is very hard to understand;
b) References are limited, please add the following and more:
Oguzlu, H., Danumah, C. and Boluk, Y., 2017. Colloidal behavior of aqueous cellulose nanocrystal suspensions. Current opinion in colloid & interface science, 29, pp.46-56.
Ye, Y., Oguzlu, H., Zhu, J., Zhu, P., Yang, P., Zhu, Y., Wan, Z., Rojas, O.J. and Jiang, F., 2023. Ultrastretchable ionogel with extreme environmental resilience through controlled hydration interactions. Advanced Functional Materials, 33(2), p.2209787.
Jiang, J., Zhu, Y., Zargar, S., Wu, J., Oguzlu, H., Baldelli, A., Yu, Z., Saddler, J., Sun, R., Tu, Q. and Jiang, F., 2021. Rapid, high-yield production of lignin-containing cellulose nanocrystals using recyclable oxalic acid dihydrate. Industrial Crops and Products, 173, p.114148.
Baldelli, A., Boraey, M.A., Oguzlu, H., Cidem, A., Rodriguez, A.P., Ong, H.X., Jiang, F., Bacca, M., Thamboo, A., Traini, D. and Pratap-Singh, A., 2022. Engineered nasal dry powder for the encapsulation of bioactive compounds. Drug Discovery Today, 27(8), pp.2300-2308.
c) Comparison with previous references is lacking.
Comments on the Quality of English LanguageNeed intense revisions.
Author Response
Point 1: English is very hard to understand
Response 1: Thank you for your kind reminder. We have carefully revised the grammar and words in the text to improve the quality of the English language. We have revised page 1, line 4, page 3, line 43, page 7, lines 157-158, page 8, lines 164-178, page 9, lines 194-201, page 10, line 222, page 11, lines 249-260, page 12, line 297, page 13, line 300, page 15, lines 336-344, page 16, lines 364-365. Lines 406-408 on page 18, 420 on page 19, and 461 on page 21 were revised. In addition, the style of references has been carefully revised.
Point 2: References are limited, please add the following and more:
Oguzlu, H., Danumah, C. and Boluk, Y., 2017. Colloidal behavior of aqueous cellulose nanocrystal suspensions. Current opinion in colloid & interface science, 29, pp.46-56.
Ye, Y., Oguzlu, H., Zhu, J., Zhu, P., Yang, P., Zhu, Y., Wan, Z., Rojas, O.J. and Jiang, F., 2023. Ultrastretchable ionogel with extreme environmental resilience through controlled hydration interactions. Advanced Functional Materials, 33(2), p.2209787.
Jiang, J., Zhu, Y., Zargar, S., Wu, J., Oguzlu, H., Baldelli, A., Yu, Z., Saddler, J., Sun, R., Tu, Q. and Jiang, F., 2021. Rapid, high-yield production of lignin-containing cellulose nanocrystals using recyclable oxalic acid dihydrate. Industrial Crops and Products, 173, p.114148.
Baldelli, A., Boraey, M.A., Oguzlu, H., Cidem, A., Rodriguez, A.P., Ong, H.X., Jiang, F., Bacca, M., Thamboo, A., Traini, D. and Pratap-Singh, A., 2022. Engineered nasal dry powder for the encapsulation of bioactive compounds. Drug Discovery Today, 27(8), pp.2300-2308.
Response 2: Thanks to your kind suggestion, we have added the above literature and added new literature on pages 3-4, lines 48-68, page 4, line 80 and page 5, lines 102-109.
Point 3: Comparison with previous references is lacking.
Response 3: Thank you for your kind suggestion, which we have analyzed against previous references on pages 14, lines 326-330, pages 17, lines 387-389, page 19, lines 425-426, and page 20, lines 439-443.
Reviewer 2 Report
Comments and Suggestions for Authors
The present work is pertaining to synthesize Self-assembled amphiphilic chitosan nanomicelles followed by its characterization and Antibacterial Activity. The present study aimed to improve both the antibacterial properties and stability of chitosan, and thus synthesized amphiphilic chitosan using the hydrophobic modification of chitosan with deoxycholic acid and hydrophilic modification of chitosan with succinic anhydride. The idea as well as the objectives of this work are quite interesting. The experimental work as well as the discussion part are clear. State of art analysis was used to characterize the end product,. So, i recommend accepting this work in its present form after some language editing.
Comments on the Quality of English LanguageNeed some language editing with respect to grammar and typo mistakes, to improve the final version.
Author Response
Question: Need some language editing with respect to grammar and typo mistakes, to improve the final version.
Response: Thank you for your kind reminder. We have carefully revised the grammar and words in the text to improve the quality of the English language. We have revised page 1, line 4, page 3, line 43, page 7, lines 157-158, page 8, lines 164-178, page 9, lines 194-201, page 10, line 222, page 11, lines 249-260, page 12, line 297, page 13, line 300, page 15, lines 336-344, page 16, lines 364-365. Lines 406-408 on page 18, 420 on page 19, and 461 on page 21 were revised. In addition, the style of references has been carefully revised.
Reviewer 3 Report
Comments and Suggestions for Authors
This manuscript deals with "Self-assembled amphiphilic chitosan nanomicelles: Synthesis, Characterization and Antibacterial Activity " I suggest a minor correction and require a detailed clarification. A correction should be addressed by the authors as follows: The abstract is not well organized; the sentences are incomplete, and there is no sense of continuity. It would be feasible if you included the significance of the current study in the abstract. A brief description of how the authors selected information from the literature in the databases, as well as what time period they searched for, is missing. The authors should justify and expand the information on the advantages of this work for biomedical applications. Authors should specify the main experimental conditions used based on the evidence from the literature. Where they briefly describe the most important data reported in the literature in a homogeneous manner and reinforce the relevance of this method as novel alternatives. Authors should discuss whether the use of work represents a solid alternative to existing therapeutics. Also, please discuss the use of method using green nanomaterials.. The article lacks some background information on the significance of studying. Providing a brief introduction or literature review on the potential sources would enhance the relevance of the study. I recommend that the authors cite and comment on other previous studies that evaluated the effect of this work. There are excellent studies that should be mentioned in the introduction
-Nasibova, Aygun. "Generation of nanoparticles in biological systems and their application prospects." Adv. Biol. Earth Sci 8 (2023): 140-146.
-Ramazanli, V. N. "EFFECT OF pH AND TEMPERATURE ON THE SYNTHESIS OF SILVER NANO PARTICLES EXTRACTED FROM OLIVE LEAF." Advances in Biology & Earth Sciences 6.2 (2021).
-Jafarova, A. F., and V. N. Ramazanli. "ANTIBACTERIAL CHARACTERISTICS OF Ag NANOPARTICLE EXTRACTED FROM OLIVE LEAF." Advances in Biology & Earth Sciences 5.3 (2020).
Author Response
Question: This manuscript deals with "Self-assembled amphiphilic chitosan nanomicelles: Synthesis, Characterization and Antibacterial Activity " I suggest a minor correction and require a detailed clarification. A correction should be addressed by the authors as follows: The abstract is not well organized; the sentences are incomplete, and there is no sense of continuity. It would be feasible if you included the significance of the current study in the abstract. A brief description of how the authors selected information from the literature in the databases, as well as what time period they searched for, is missing. The authors should justify and expand the information on the advantages of this work for biomedical applications. Authors should specify the main experimental conditions used based on the evidence from the literature. Where they briefly describe the most important data reported in the literature in a homogeneous manner and reinforce the relevance of this method as novel alternatives. Authors should discuss whether the use of work represents a solid alternative to existing therapeutics. Also, please discuss the use of method using green nanomaterials. The article lacks some background information on the significance of studying. Providing a brief introduction or literature review on the potential sources would enhance the relevance of the study. I recommend that the authors cite and comment on other previous studies that evaluated the effect of this work. There are excellent studies that should be mentioned in the introduction
-Nasibova, Aygun. "Generation of nanoparticles in biological systems and their application prospects." Adv. Biol. Earth Sci 8 (2023): 140-146.
-Ramazanli, V. N. "EFFECT OF pH AND TEMPERATURE ON THE SYNTHESIS OF SILVER NANO PARTICLES EXTRACTED FROM OLIVE LEAF." Advances in Biology & Earth Sciences 6.2 (2021).
-Jafarova, A. F., and V. N. Ramazanli. "ANTIBACTERIAL CHARACTERISTICS OF Ag NANOPARTICLE EXTRACTED FROM OLIVE LEAF." Advances in Biology & Earth Sciences 5.3 (2020).
Response: Thank you for your kind suggestion, we have rewritten the abstract on page 2 to make it flow better. Also we have cited the above literature as background information for this study on page 3-4, lines 48-64 to demonstrate the strength of this work.